**Data Availability Statement:** Raw data were generated from the National Health Insurance Research Database (NHIRD) published by the Taiwan National Health Insurance (NHI) Bureau.

# Association between vesicoureteral reflux, urinary tract infection and antibiotics exposure in infancy and risk of childhood asthma

Yu-Lung Hsu[1], Cheng-Li Lin[2,3], Chang-Ching Wei [4,5]*

**1** Division of Pediatric Infectious Diseases, Department of Pediatrics, Children's Hospital, China Medical University Hospital, Taichung, Taiwan, **2** Management Office for Health Data, China Medical University Hospital, Taichung, Taiwan, **3** College of Public Health, China Medical University, Taichung, Taiwan, **4** Division of Pediatric Allergy, Immunology, and Rheumatology, Department of Pediatrics, Children's Hospital, China Medical University Hospital, Taichung, Taiwan, **5** School of Medicine, China Medical University, Taichung, Taiwan

* weilonger@gmail.com

## Abstract

### Background

The use of antibiotics for treating infection in childhood and their association with increased risk of asthma remain controversial. Infants diagnosed with vesicoureteral reflux (VUR) belong to a unique population who are administered antibiotics for a long time and are susceptible to recurrent UTI. It is interesting to study the risk of asthma in these infants with or without VUR.

### Methods

Taiwanese children born between 2000 and 2007 were enrolled in population-based birth cohort study. Participants diagnosed with VUR and UTI within first year were classified into four groups (VUR, UTI, VUR and UTI, and control). We calculated follow-up person-years for each participant from the index date until the asthma diagnosis, their withdrawal from the insurance system (because of death or loss to follow-up), or till the end of 2008. The risk of asthma was compared between the 4 cohorts by using Cox proportional hazards model analysis, adjusted hazard ratio (aHR), and 95% confidence interval (95% CI).

### Results

Children diagnosed with VUR (n = 350), UTI (n = 15542), VUR and UTI (n = 1696), and randomly selected controls (n = 17588) were enrolled. The overall rate of incidence of asthma was found to be 1.64-fold, 1.45-fold, and 1.17-fold higher in the UTI, VUR/UTI, and VUR cohorts than in the controls (5.60, 5.07, and 4.10 vs. 3.17 per 100 person-years), respectively. After adjusting the potential factors, the overall risk of asthma remained the highest in UTI (aHR: 1.74, 95% CI : 1.65 to 1.80) followed by VUR/UTI (aHR: 1.56, 95% CI :

**Funding:** This study is supported in part by Clinical Trial Center and Department of Chinese Medicine and Pharmacy, Ministry of Health and Welfare (MOHW109-TDU-B-212-114004), China Medical University Hospital (DMR-HHC-110-7). The funders had no role in study design, data collection and analysis, decision to publish, or preparation of the manuscript.

**Competing interests:** No authors have competing interests.

1.40 to 1.75) and VUR cohorts (aHR: 1.25, 95% CI: 0.96 to 1.62). The incidence of asthma was higher in boys than in girls.

## Conclusion

The nationwide retrospective cohort study demonstrated that short-term therapeutic dose of antibiotics for UTI in infants with or without VUR has a positive correlation with the prevalence of childhood asthma. Significant risk of childhood asthma was not observed when VUR cohort was exposed to long-term low-dose of prophylactic antibiotics.

## Introduction

Despite recent advances in understanding the pathophysiology of childhood asthma, its incidence has been increasing rapidly worldwide over the past 2 decades [1–3]. Although research has investigated possible explanations for this increase, underlying etiologies have not been fully elucidated and therefore remain controversial [4]. Asthma has been considered as a chronic inflammation of the airways resulting from a complex interplay between genetic and environmental factors [4]. Although genetic predisposition for its prevalence is clearly evident, gene-by-environment interactions probably explain much of the international variation and global increase [5,6]. Accumulating evidence indicates that early life events such as preterm birth, low birth weight, respiratory viral infections, use of antibiotics and acetaminophen may play a role in its development [5–14].

Urinary tract infection (UTI) is a common bacterial disease in childhood [15]. Common antibiotic treatment is promptly initiated to eradicate febrile UTI, prevent bacteremia, and reduce the risk and extent of renal parenchymal injury in infants and young children [15]. VUR is the retrograde flow of urine from the bladder to the ureter and kidneys, and is a predisposing factor for pyelonephritis and renal scarring [16]. Approximately 70% of all the infants who are diagnosed with recurrent febrile UTI, will have VUR [17]. Once VUR has been diagnosed, the basic premise in management is to prevent further ascending UTI which would lead to potential renal damage. The interventions include observation, continuous antibiotic prophylaxis (CAP), surgery or a combination of both [18]. Since 1960s, antibiotic prophylaxis has been one of the management options in treating VUR to reduce the recurrence rate of UTI in children with VUR, especially in children under age five who may be more susceptible to renal damage by ascending UTI. Since reflux will resolve over a period of time in significant proportion of children, CAP remains the mainstay of initial management of infantile VUR to prevent recurrent UTI [16–18]. Actually, some children with low grade VUR may take CAP for several years until such time that the reflux would disappear. To date, most studies concern the issue of long-term antibiotic prophylaxis about the efficiency of reducing frequency of UTI and prevention of renal damage, and antimicrobial resistance with breakthrough infection, but no study investigate whether the use of antibiotic prophylaxis in the management of VUR might be associated with the risk of allergic diseases.

Concept of antibiotic use in early life is associated with the development of later allergic diseases is well-known in literature, but how subsequent risk of allergic diseases is relating to long-term CAP remains unclear. To date, no RCT or real-life observation study have been performed to examine the difference between long-term continuous low dose antibiotics and short course therapeutic antibiotics on risks of childhood asthma [10–13]. From the literature, most studies address short-term postnatal antibiotics can have significant effects on the

evolution of the infant gut microbiota, the long-term health implications of which remain unknown. One observational study found an increased subsequent risk of asthma in children who were diagnosed with neonatal UTI [19]. However, in this study, children who were diagnosed with VUR were excluded from the study and the role of antibiotics or infection in the development of asthma remained unclear.

Even through the emerging evidence associated early life events in the development of asthma, the role of infantile VUR with CAP and/or UTI remain unclear. A population-based birth cohort study investigated the incidence and risk of asthma in 4 groups of children had been diagnosed with VUR and/or UTI and control during infancy.

## Methods

### Data sources

Taiwan launched a single-payer compulsory National health insurance (NHI) program at the beginning of 1995 that covered almost all citizens in a modest cost [20,21]. All the data related to health claims were collected in the National health insurance research database (NHIRD) and managed by the NHI program (http://www.nhi.gov.tw/english/index.aspx) [20–22]. The file derived from NHIR contained a randomly selected sample of half the population of Taiwanese children (age < 18 years) [23]. The disease was coded based on the international classification of diseases, 9th revision, and clinical modification (ICD-9-CM). Identities of the insured children were encrypted and all data were analyzed anonymously to comply with the Personal Information Protection act. The study was approved by the Institutional Review Board of China Medical University Hospital (CRREC-103-048). All methods were performed in accordance with the relevant guidelines. The disease was coded based on the ICD-9-CM.

### Study design and participants

The outcome of this study was the diagnosis of asthma. Participants who were born during 2000–2007 were enrolled in 4 birth cohorts based on their diagnosis: (1) VUR cohort: children who had been diagnosed with VUR (ICD-9-CM codes 593.7) within first year but had never been diagnosed with UTI (ICD-9-CM codes 599.0) (2) UTI cohort: children who had been diagnosed with UTI within first year but had never been diagnosed with VUR (3) VUR and UTI (VUR/UTI) cohort: children who had been diagnosed with VUR and UTI within first year (4) control cohort: infants who had never been diagnosed with UTI and VUR. In this study, VUR only cohort is a group of children who diagnosed as VUR within first year of age, is too young to do surgical intervention to correct VUR. Therefore, they usually take long-term prophylactic antibiotics. Hence, VUR group is a surrogate of participant on long-term prophylactic antibiotics. The date of diagnosis of the three study groups was considered as the baseline (index date). The method of 1:1 matching was used to screen healthy participants to control potential confounders. An infant who was not diagnosed with UTI and VUR was randomly selected. Each infant of the study groups was matched according to sex, age (in one-year intervals), urbanization level of residual area, parental occupation, and baseline year. We classified urbanization into 7 ordered levels according to population density (people/km$^2$), proportion of people with higher education, size of elderly and agricultural population, and number of physicians per 100,000 people. Due the relatively small number of people in the levels 5, 6, and 7 we combined them with level 4. Level 1 was most urbanized whereas level 1 was least urbanized. Asthma was defined in children having symptoms related to at least 3 ICD-9-CM code and 493 records in field of inpatient or ambulatory care claimed to improve the diagnostic accuracy to avoid overestimation of incidence of asthma. Participants having a history of asthma before baseline or missing demographic information were excluded. Preterm

children, low birth weight (ICD-9-CM: 764 and 765), respiratory conditions related to fetus and newborns (ICD-9-CM: 770), neurogenic bladder (ICD-9-CM code 344.61), cauda equina syndrome along with neurogenic bladder (ICD-9-CM code 330), and congenital urinary tract anomaly (ICD-9-CM code 753) were also excluded. Comorbidities included upper and lower respiratory tract infections (ICD-9-CM code 460–5 and ICD-9-CM code 466, 480–6), suppurative otitis media (ICD-9-CM code 382), and acute sinusitis (ICD-9-CM code 461). All participants followed up until the asthma diagnosis, their withdrawal from the insurance system (because of death or loss to follow-up), or till the end of 2008.

## Statistical analysis

Person-years was used as the denominator to estimate the rate of incidence for follow-up of an individual during the duration of the study. Follow-up person-years was calculated starting from the index date until the date of the diagnosis or was censored due to lack of follow-up, death or withdrawal from the insurance system until the end of 2008. Follow-up person-years was used to estimate the incidence density rates (IR) and hazard ratios (HRs) of asthma for the 3 study cohorts that were compared with those of the control cohort that was adjusted for age, sex, urbanization, and comorbidities. HR was determined using the Cox proportional hazard regression model. Kaplan-Meier plots showed the cumulative incidence of asthma and the log-rank test was used to measure the difference between the study and the control cohorts. All data analyses were performed using SAS statistical software, version 9.2, for Windows (SAS Institute Inc., Cary, NC). Kaplan-Meier survival curves were plotted using R software, version 2.14.1 (R Development Core Team, Vienna, Austria). A two-sided $p$ value < 0.05 was considered statistically significant.

## Results

In this study, 350,15542,1696, and 17588 children participated in the VUR, UTI, VUR/UTI, and control cohort, respectively (Table 1). The mean ages (SD) were 5.2 (3.09) and 5.95 (3.48) years in study and control cohorts, respectively. The SDs of the follow-up year were 4.18 (2.42) and 4.53 (2.39) in the study cohorts and control cohort, respectively. The majority of the participants were boys (57.2%). The occurrence of most respiratory tract infections (upper, lower respiratory tract infections, and suppurative otitis media) was significantly higher in the UTI cohorts followed by the VUR/UTI cohort and the control cohort; the occurrence was lowest in the VUR cohort ($p$ < 0.001) (Table 1). The overall antibiotics use for any cause during study period were shown in both Tables 1 and 2. The highest use of 1st first-generation cephalosporin was in the VUR cohort followed by the VUR/UTI, UTI, and control cohorts. Third generation cephalosporins and aminoglycosides were used in the highest frequency in the VUR/UTI cohort whereas they were used in the least frequency in the control cohort (Table 1). Penicillin based antibiotics was used in the highest frequency in the UTI cohort whereas it was used in the least frequency in the control cohort (Table 1). Table 2 demonstrated health care and antibiotics utilization among children with vesicoureteral reflux (VUR) and urinary tract infection (UTI). Children with UTI and VUR had the highest rate of more medical visits, admission days, and days of antibiotics use; they had more frequency to use 1st and 3rd generation cephalosporin, aminoglycoside and penicillin (Table 2). Table 3 demonstrated the risk of asthma compared to children with UTI and VCUG. The incidence of asthma was 1.64-fold higher in the UTI cohort as compared with the control cohort (5.60 vs. 3.17 per 10,000 person-years). It was 1.45-fold higher in the VUR/UTI cohort as compared with the control cohort (5.07 vs. 3.17 per 10,000 person-years) (Table 3). The adjusted HR for asthma was greater in children older than 6 months (UTI cohort: 1.94 (95% CI: 1.79–2.10), VUR/UTI cohort 1.83 (95% CI:

**Table 1. Demographics among the 4 cohorts: Vesicoureteral reflux (VUR) cohort, urinary tract infection (UTI) cohort, VUR and UTI cohort, and control cohort.**

| | Control (N = 17588) | | Total (N = 17588) | | VUR only (N = 350) | | UTI only (N = 15542) | | VUR and UTI (N = 1696) | | p-value |
|---|---|---|---|---|---|---|---|---|---|---|---|
| | n | (%) | n | (%) | n | (%) | n | (%) | n | (%) | |
| Age, months, mean (SD)* | 5.95 | (3.48) | 5.20 | (3.09) | 3.79 | (2.68) | 5.32 | (3.12) | 4.33 | (2.70) | <0.001 |
| Follow-up time, years, mean (SD)* | 4.53 | (2.39) | 4.18 | (2.42) | 4.11 | (2.38) | 4.16 | (2.42) | 4.29 | (2.36) | <0.001 |
| Sex | | | | | | | | | | | 0.98 |
| Girl | 7528 | (42.8) | 7530 | (42.8) | 86 | (24.6) | 6854 | (44.1) | 590 | (34.8) | |
| Boy | 10060 | (57.2) | 10058 | (57.2) | 264 | (75.4) | 8688 | (55.9) | 1106 | (65.2) | |
| Comorbidity | | | | | | | | | | | |
| URI | 10876 | (61.8) | 12975 | (73.8) | 195 | (55.7) | 11599 | (74.6) | 1181 | (69.6) | <0.001 |
| LRI | 5310 | (30.2) | 6851 | (39.0) | 93 | (26.6) | 6233 | (40.1) | 525 | (31.0) | <0.001 |
| Otitis media | 429 | (2.44) | 736 | (4.18) | 7 | (2.00) | 690 | (4.44) | 39 | (2.30) | <0.001 |
| Sinusitis | 2168 | (12.3) | 2117 | (12.0) | 24 | (6.86) | 1936 | (12.5) | 157 | (9.26) | 0.41 |
| Use of antibiotics | | | | | | | | | | | |
| Cephalosporin | | | | | | | | | | | |
| 1st generation | 1249 | (7.10) | 2195 | (12.5) | 58 | (16.6) | 1876 | (12.1) | 261 | (15.4) | <0.001 |
| 2nd generation | 303 | (1.72) | 316 | (1.80) | 5 | (1.43) | 287 | (1.85) | 24 | (1.42) | 0.60 |
| 3rd generation | 2 | (0.01) | 28 | (0.16) | 0 | (0.00) | 22 | (0.14) | 6 | (0.35) | <0.001 |
| Aminoglycoside | 43 | (0.24) | 361 | (2.05) | 7 | (2.00) | 309 | (1.99) | 45 | (2.65) | <0.001 |
| Penicillins | 3545 | (20.2) | 4599 | (26.2) | 82 | (23.4) | 4092 | (26.3) | 425 | (25.1) | <0.001 |

Chi-square test

&Fisher-exact test, and

*t-test comparing subjects with and without VCUG.

Abbreviations: URI, upper respiratory tract infections; LRI, low respiratory tract infections.

1.49–2.25), in girls (UTI cohort: 1.85 (95% CI: 1.70–2.02), and VUR and UTI cohort 1.75 (95% CI: 1.45–2.12) (Table 3). Moreover, the adjusted HR was greater in children who were diagnosed with respiratory tract infections, suppurative otitis media, and sinusitis in both the UTI and VUR/UTI cohorts (Table 3). Table 4 demonstrated the incidence, crude and adjusted hazard ratio of asthma between VUR patients with and without specific treatment. Children with VUR only received reimplantation had lower risk for asthma (Table 4). Children with VUR

**Table 2. Health care and antibiotics utilization among children in 3 study groups: Vesicoureteral reflux (VUR), urinary tract infection (UTI) and VUR & UTI.**

| | Total (N = 17588) | | VUR only (N = 350) | | UTI only (N = 15542) | | VUR & UTI (N = 1696) | | p-value |
|---|---|---|---|---|---|---|---|---|---|
| | mean | (SD) | mean | (SD) | mean | (SD) | mean | (SD) | |
| Frequency of medical visit, mean (SD) | 3.41 | (6.31) | 0.42 | (0.72) | 2.52 | (3.79) | 12.2 | (14.0) | <0.001 |
| Admission days for UTI | 95.8 | (430.8) | 14.6 | (64.1) | 84.7 | (379.2) | 214.2 | (767.8) | <0.001 |
| Mean days of use antibiotics, mean (SD), (day) | | | | | | | | | |
| Cephalosporin | | | | | | | | | |
| 1st generation | 8.05 | (15.8) | 6.57 | (5.68) | 7.50 | (11.6) | 12.3 | (33.3) | <0.001 |
| 2nd generation | 7.51 | (7.52) | 10.8 | (10.9) | 7.55 | (7.66) | 6.42 | (4.45) | 0.48 |
| 3rd generation | 11.1 | (13.6) | 0 | (0.00) | 10.9 | (14.1) | 12.0 | (12.8) | 0.87 |
| Aminoglycoside | 4.40 | (4.27) | 3.14 | (2.19) | 4.32 | (4.35) | 5.13 | (3.88) | 0.36 |
| Penicillin | 11.1 | (15.5) | 12.1 | (15.2) | 10.9 | (15.1) | 13.0 | (19.1) | 0.02 |

one way ANOVA comparing among three groups.

**Table 3. The risk of asthma among children with vesicoureteral reflux (VUR), urinary tract infection (UTI) and VUR and UTI compared to control stratified by demographics in Cox proportional hazard regression.**

| | Control | | VUR only | | Adjusted HR[†] (95% CI) | UTI only | | Adjusted HR[†] (95% CI) | VUR and UTI | | Adjusted HR[†] (95% CI) |
|---|---|---|---|---|---|---|---|---|---|---|---|
| | Event | IR | Event | IR | | Event | IR | | Event | IR | |
| All | 2529 | 3.17 | 59 | 4.10 | 1.25(0.96, 1.62) | 3625 | 5.60 | 1.74(1.65, 1.83)** | 369 | 5.07 | 1.56(1.40, 1.75)** |
| Stratified age | | | | | | | | | | | |
| ≦6 months | 1364 | 2.97 | 47 | 4.05 | 1.19(0.89, 1.59) | 2172 | 5.54 | 1.63(1.52, 1.75)** | 270 | 5.02 | 1.48(1.29, 1.68)** |
| >6 months | 1165 | 4.05 | 12 | 4.33 | 1.47(0.83, 2.60) | 1453 | 5.69 | 1.94(1.79, 2.10)** | 99 | 5.22 | 1.83(1.49, 2.25)** |
| Sex | | | | | | | | | | | |
| Girl | 928 | 2.66 | 12 | 3.37 | 1.38(0.78, 2.45) | 1464 | 5.01 | 1.85(1.70, 2.02)** | 121 | 4.59 | 1.75(1.45, 2.12)** |
| Boy | 1601 | 3.58 | 47 | 4.34 | 1.18(0.88, 1.58) | 2161 | 6.09 | 1.65(1.54, 1.77)** | 248 | 5.35 | 1.46(1.28, 1.68)** |
| Comorbidiy | | | | | | | | | | | |
| No | 841 | 3.02 | 20 | 3.35 | 1.09(0.70, 1.69) | 721 | 5.20 | 1.74(1.58, 1.93)** | 76 | 3.93 | 1.30(1.02, 1.64)* |
| Yes | 1688 | 3.26 | 39 | 4.64 | 1.44(1.04, 1.98)* | 2904 | 5.71 | 1.73(1.63, 1.84)** | 293 | 5.48 | 1.69(1.49, 1.92)** |

IR, incidence rate, per 1,00 person-years; HR: Hazard ratio.

[†]The urbanization level was categorized by the population density of the residential area into 4 levels, with level 1 as the most urbanized and level 4 as the least urbanized.

Adjusted HR[†], adjusted for age, sex, urbanization level, and parental occupation, type of antibiotic administration, commodities, types of treatment for VUR, and comorbidities of URI, LRI, otitis media, and sinusitis.

* p<0.05

**p<0.001.

and UTI received any procedure, such as reimplantation and subtrigonal injection had lower risk for asthma (Table 4). The Kaplan–Meier survival analysis in the Fig 1 showed that the cumulative incidence of asthma from the highest to lowest were in VUR/UTI, UTI, VUR, and control cohorts.

## Discussion

We revealed the association of VUR and UTI with the risk of development of asthma during the infantile period. In the previous study, although neonatal UTI was found to increase the subsequent risk of asthma in children, the potential comorbidity of VUR was not considered [19]. Therefore, this is the first retrospective population- based cohort study that discusses the association of VUR and infantile UTI with the risk of development of childhood asthma.

The UTI cohort was compared with the control cohort followed by the VUR/UTI cohort that had a significantly adjusted HR for childhood asthma than the VUR cohort. This finding indicated that infantile UTI had an influence on childhood asthma. Lin et al. first used a population-based cohort study to demonstrate that neonatal UTI increases the risk of childhood asthma [19]. It can be inferred from the results that neonatal and infantile UTIs contribute to the risk of childhood asthma. In a previous study, researchers explained the contribution of urinary infection to childhood asthma. Microorganisms responsible for UTIs influence innate immunity through toll-like receptor signal transduction pathway that trigger impaired type 1 helper T cells inducing a sensitization reaction of asthma [19].

UTI cohorts had significantly higher occurrences of respiratory tract infections (upper, lower respiratory tract infections, and suppurative otitis media)followed by the VUR/UTI and the control cohort. Occurrence of these infections were lowest in the VUR cohort. UTI and VUR/UTI cohorts had a greater adjusted HR for childhood asthma when they were combined with respiratory tract infections, suppurative otitis media, and sinusitis. Respiratory tract infection caused by the respiratory syncytial virus (RSV) in childhood has been documented to

**Table 4. Incidence, crude and adjusted hazard ratio of asthma among children with vesicoureteral reflux (VUR) stratified by specific treatments for VUR in Cox proportional hazard regression.**

| Therapy for VUR | N | Event | Person-years | IR | Crude HR * (95% CI) | Adjusted HR† (95% CI) |
|---|---|---|---|---|---|---|
| **VUR only** | | | | | | |
| (1) Reimplantation | | | | | | |
| No | 294 | 51 | 1198 | 4.26 | 1.00(Reference) | 1.00(Reference) |
| Yes | 56 | 8 | 240 | 3.33 | 0.78(0.37, 1.64) | 0.85(0.40, 1.81) |
| (2) Antireflux procedure with subtrigonal injection | | | | | | |
| No | 334 | 56 | 1372 | 4.08 | 1.00(Reference) | 1.00(Reference) |
| Yes | 16 | 3 | 66 | 4.56 | 1.17(0.37, 3.73) | 1.68(0.51, 5.55) |
| (1)+(2) | | | | | | |
| No | 349 | 59 | 1433 | 4.12 | 1.00(Reference) | 1.00(Reference) |
| Yes | 1 | 0 | 5 | 0.00 | - | - |
| **VUR and UTI** | | | | | | |
| (1) Reimplantation | | | | | | |
| No | 1282 | 275 | 5419 | 5.07 | 1.00(Reference) | 1.00(Reference) |
| Yes | 414 | 94 | 1856 | 5.07 | 1.02(0.81, 1.29) | 1.05(0.83, 1.32) |
| (2) Antireflux procedure with subtrigonal injection | | | | | | |
| No | 1567 | 349 | 6738 | 5.18 | 1.00(Reference) | 1.00(Reference) |
| Yes | 129 | 20 | 537 | 3.73 | 0.69(0.44, 1.09) | 0.70(0.45, 1.11) |
| (1)+(2) | | | | | | |
| No | 1677 | 367 | 7176 | 5.11 | 1.00(Reference) | 1.00(Reference) |
| Yes | 19 | 2 | 99 | 2.03 | 0.41(0.10, 1.63) | 0.39(0.11, 1.56) |

IR, incidence rate, per 1000 person-years.

Crude HR*, relative hazard ratio.

Adjusted HR†, adjusted for age, sex, urbanization level, and parental occupation, type of antibiotic administration, commodities, and comorbidities of URI, LRI, otitis media, and sinusitis.

increase the risk of asthma later [4]. Consistent with our findings, children diagnosed with infantile UTI and respiratory tract infections had a higher incidence of childhood asthma. Respiratory tract infections, suppurative otitis media, and sinusitis also increased the dependency on antibiotics. Meta-analysis studies [4,12,24] reported significantly increasing risk of asthma with the use of antibiotics in the infantile period. The possible mechanism for this association is that administration of antibiotics in infancy causes change in the intestinal or airway microbiome during the import period of initial immune development to induce childhood asthma [25]. However, there was high heterogeneity across the meta-analysis studies. Moreover, the association between antibiotic use in the first year of life and childhood asthma became insignificant only when prospective studies were analyzed. Hoskin-Parr et al. and Pitter et al. reported a dose-response relationship between antibiotic use in early life and childhood asthma [26,27]. The risk of childhood asthma increased with the number of antibiotic courses in infancy. However, these studies did not identify further risk of asthma between short-term and long-term antibiotic use. Mild and moderate VUR is expected to resolve spontaneously in most children who are treated medically. Surgical correction is usually recommended when children are older than one year. From 2011, American academy of pediatrics guidelines did not support the use of antimicrobial prophylaxis to prevent UTI in infants diagnosed with VUR [18]. As period of study was before the revised guidelines, antibiotic prophylaxis was prescribed regularly for Taiwanese infants who were diagnosed with VUR. The VUR cohort that was exposed to long-term low-dose prophylactic antibiotics had no statistically

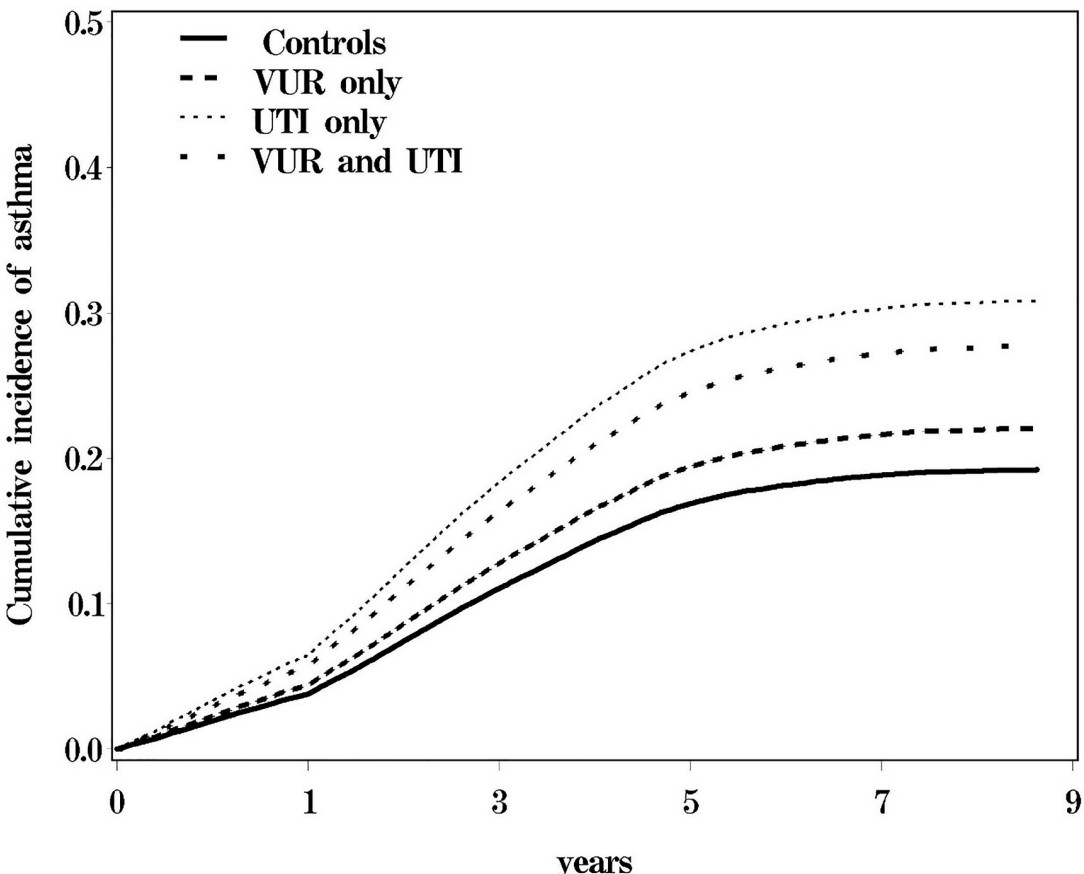

**Fig 1. The Kaplan-Meier analysis of cumulative incidence of asthma among children with vesicoureteral reflux (VUR), urinary tract infection (UTI) and VUR and UTI compared to control cohort.**

significant risk of childhood asthma. Although this risk was significantly higher in the UTI cohort followed by the VUR/UTI cohort, it was not present in VUR cohort as it was not observed to have inflamed UTI when antibiotics were administered. The VUR cohort was exposed to low-dose long-term prophylactic antibiotics but the other two cohorts experienced short-term high-dose therapeutic antibiotics. This finding implied that long-term administration of antibiotics in lower doses might not influence childhood asthma. UTI cohort and VUR/UTI cohort had a significantly higher risk of childhood asthma than the VUR cohort. Administration of high dose therapeutic antibiotics for a short term in early life and infantile UTI play an important role in influencing childhood asthma.

The findings were inconsistent with the hygiene hypothesis. In 1989, Strachan introduced the original hygiene hypothesis that stated that the prevalence of atopic disease was lower due to larger household size in a relatively unhygienic environment [28]. This hypothesis was augmented to state that increasing exposure to viral and bacterial infections in childhood would have some protective effects against asthma [29]. However, review articles also pointed out that not all childhood infections have an effect in reducing the risk of asthma [25,29].

In this study, infants > 6 months had a higher risk of childhood asthma in three cohorts. Örtqvist et al. reported a decreased association between administration of antibiotics for urinary tract/skin infections and subsequent development of asthma with an increase in age from 0 to 2 years. This excess risk disappeared after sibling analyses were adjusted [30]. Therefore,

sibling factor may be considered as a confounder for the finding. In addition to the hygiene hypothesis, infantile intestinal and airway microbiomes contribute further to risk of asthma [25]. The changes in the intestinal and airway microbiomes were not observed after infantile UTI was treated with antibiotics. Further clinical prospective studies are needed to study the link between microbiome and asthma risk. Although predominance of males was observed to be a factor that was same as that in our study for the prevalence [31], female infants belonging to the three cohorts had a higher risk than the control cohort. Normann et al. found that *Chlamydia pneumoniae* infection in childhood may be a significant risk factor for asthma in girls. This difference might be related to the susceptibility and less ability of immune responses to eliminate *Chlamydia pneumoniae*[32]. However, the current study design could not associate any immune responses with infantile UTI. Further longitudinal studies are therefore needed to determine the relationship between gender differences in eliminating the UTI and childhood asthma.

Managements for primary VUR include observation, antibiotic prophylaxis, and surgery. These procedures include open surgical reimplantation (ureteroneocystostomy) and endoscopic correction (anti-reflux procedure along with subtrigonal injection). Surgical intervention is indicated for patients who are under antimicrobial prophylaxis or high-grade reflux for persistent VUR and recurrent UTI. Data showed that the risk of childhood asthma can be reduced by performing an anti-reflux procedure along with a subtrigonal injection in the VUR/UTI cohort. Ureteroneocystostomy in the VUR and VUR/UTI cohorts can also decrease the risk of childhood asthma. Open surgical reimplantation as well as endoscopic correction can reduce the occurrence of febrile UTI or acute pyelonephritis [33,34] reducing the risk of childhood asthma. It was observed that anti-reflux procedure along with subtrigonal injection could not reduce the risk of asthma in the VUR cohort; therefore, it can be implied that an open surgery has a higher success rate then endoscopic correction.

There are some limitations with the present study. Firstly, VUR could not be graded based on disease codes; therefore, an association between the severity of VUR and asthma could not be identified. Secondly, detailed clinical presentation and laboratory data for UTI such as white blood cell count, C-reactive protein, pathogenicity of urine culture, severity and frequency of UTI were lacking. Therefore, the association between asthma and the parameters of the UTI cannot be discussed. Thirdly, as the detailed profile related to the course and use of antibiotics was absent in this study; their influence on occurrence of asthma cannot be indicated. Fourthly, we did not analyze participants had UTI or VUR after the first year of life because the aim of our study is to early-life events and antibiotics use might contribute to the development of allergic diseases in later life. In addition, VUR is found in 30–45% of children presenting with a febrile UTI, with an even higher risk in neonates.[17,18] Hence, our study is focus on whether UTI, VUR, or VUR+UTI and antibiotics exposure and treatment for VUR within first year are really risk factors for childhood asthma.

## Conclusions

Short-term therapeutic dose of antibiotics for UTI in infants with or without VUR has a positive correlation with the prevalence of childhood asthma. However, significant risk of childhood asthma was not observed when VUR cohort was exposed to long-term low-dose of prophylactic antibiotics implying that they have less influence on its occurrence.

## Author Contributions

**Conceptualization:** Chang-Ching Wei.

**Funding acquisition:** Yu-Lung Hsu.

**Investigation:** Yu-Lung Hsu.

**Methodology:** Yu-Lung Hsu, Cheng-Li Lin.

**Project administration:** Cheng-Li Lin, Chang-Ching Wei.

**Software:** Cheng-Li Lin.

**Writing – original draft:** Chang-Ching Wei.

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
