## [Decision Letter · Decision Letter 0]

26 Nov 2020

PONE-D-20-27348

Dose continuous antibiotic prophylaxis for vesicoureteral reflux increase the subsequent risk of childhood asthma?

PLOS ONE

Dear Dr. Wei,

Thank you for submitting your manuscript to PLOS ONE. After careful consideration, we feel that it has merit but does not fully meet PLOS ONE’s publication criteria as it currently stands. Therefore, we invite you to submit a revised version of the manuscript that addresses the points raised during the review process.

We look forward to receiving your revised manuscript.

Kind regards,

Man Ki Kwok

Academic Editor

PLOS ONE

Journal Requirements:

2. In your ethics statement in the Methods section and in the online submission form, please provide additional information about the data used in your retrospective study.

Specifically, please ensure that you have discussed whether all data were fully anonymized before you accessed them and/or whether the IRB or ethics committee waived the requirement for informed consent.

3. Please include the date(s) on which you accessed the databases or records to obtain the data used in your study.

'This study is supported in part by Clinical Trial Center and Department of Chinese Medicine and Pharmacy, Ministry of Health and Welfare (MOHW109-TDU-B-212-114004), China Medical University Hospital (CRS-108-015, DMR-HHC-109-9, and DMR-108-200).'

'NO - Include this sentence at the end of your statement: The funders had no role in study design, data collection and analysis, decision to publish, or preparation of the manuscript.'

Reviewers' comments:

Reviewer's Responses to Questions

**Comments to the Author**

1. Is the manuscript technically sound, and do the data support the conclusions?

Reviewer #1: Partly

Reviewer #2: Partly

2. Has the statistical analysis been performed appropriately and rigorously? 

Reviewer #1: Yes

Reviewer #2: No

3. Have the authors made all data underlying the findings in their manuscript fully available?

Reviewer #1: Yes

Reviewer #2: No

4. Is the manuscript presented in an intelligible fashion and written in standard English?

Reviewer #1: Yes

Reviewer #2: Yes

5. Review Comments to the Author

Reviewer #1: The review for the paper entitled ‘Dose continuous antibiotic prophylaxis for vesicoureteral reflux increase the subsequent risk of childhood asthma?’ has been completed. The authors concluded that the nationwide retrospective cohort study in Taiwan demonstrated that neonatal urinary tract infection (UTI) may have a greater impact on the development of childhood asthma.

However, this paper does not seem to be suitable for publication because of the following reasons:

1) The concept that antibiotic use for various infectious diseases including UTI in early life is associated with the development of later allergic diseases is not novel and there are many articles in this concept (Tanaka M, Nakayama J. Development of the gut microbiota in infancy and its impact on health in later life. Allergol Int. 2017 Oct;66(4):515-522; Gensollen T, Iyer SS, Kasper DL, Blumberg RS. How colonization by microbiota in early life shapes the immune system. Science. 2016 Apr 29;352(6285):539-44; Lin CH, Lin WC, Wang YC, Lin IC, Kao CH. Association Between Neonatal Urinary Tract Infection and Risk of Childhood Allergic Rhinitis. Medicine (Baltimore). 2015 Sep;94(38):e1625; Lin CH, Wang YC, Lin WC, Kao CH. Neonatal urinary tract infection may increase the risk of childhood asthma. Eur J Clin Microbiol Infect Dis. 2015 Sep;34(9):1773-8)

2) They speculate the association of the history of neonatal UTI with an asthma in later life is caused by dysbiosis of gut microbiota due to antibiotic use in infancy. If so, they should explore the dysbiosis of gut microbiota in neonates with UTI. Without such information, their speculation cannot be supported as numerous confounding factors exist.

Reviewer #2: This is a very interesting article on the impact of antibiotic prophylaxis for vesicoureteral reflux and the subsequent risk of childhood asthma. The authors included a cohort of children from a nationwide database that were diagnosed with either VUR, UTI, VUR+UTI, and a control group during the first year of life, and investigated the risk of asthma in each of the 4 groups, with the primary endpoint being diagnosed with asthma. The authors found that the overall incidence of asthma was highest in the UTI group, and the incidence of asthma was higher in boys than girls.

Major comments:

1) Although the author's primary aim of this study is very interesting and the authors have put good efforts to investigate, I believe the statistical analyses which the authors have used to investigate -- comparing the risk of asthma in the selected cohort of patients and finding the HR compared with the control group---is not enough to explain that there is a higher risk of asthma in children with UTI or UTI+VUR. The reason for this is because there are many confounding factors which the authors have not fully adjusted for. I recommend that the authors perform at least a multivariate analyses to find whether UTI, VUR, or VUR+UTI are really risk factors for asthma by adjusting for confounding such as sex, previous antibiotic administration, commodities, types of treatment for VUR...etc. If this is not possible, I believe that the authors can only conclude that there is a higher incidence of asthma in these patient groups; however, whether UTI or VUR or UTI/VUR are risk factors for asthma cannot not be concluded from the data presented in this study.

2) The title states that continuous antibiotic prophylaxis for VUR increases the risk of asthma, however, the results and discussion do not support this statement. In fact, the data presented by the authors show that UTI has a significantly higher HR for childhood asthma, and explain that the reason for this may be due to the microorganisms responsible for UTIs which influence the innate immunity triggering the sensitization of asthma. I suggest that the authors change the title to better fit the findings of the study.

3) The UTI cohort was found to have significantly higher occurrence of respiratory tract infections compared to the VUR/UTI and control cohort. The risk of asthma was highest in the URI cohort, followed by VUR/UTI cohort. Again, with the data that the authors have presented, it is unclear whether UTI is associated with a higher risk of asthma after adjusting for respiratory tract infections.

4) In the study participants, patients were included based on whether they had UTI or VUR during the first year of life. Information about whether the study participants had UTI or VUR after the first year of life is important.

5) Table 1 and 2 both have "use of antibiotics". Does table 2 show the type of antibiotics used only to treat UTI's? Did the UTI's occur after the 1st year of life? of after their diagnosis of VUR? These points need to be clarified.

6) For patients without the diagnosis of asthma, what was the age until follow up?

7) In the discussion, line 286-288, it is written that "The VUR cohort was exposed to low-dose long-term prophylactic antibiotics but the other two cohorts experienced short-term high-dose therapeutic antibiotics". However, this data is not presented in the results. Data needs to be given on how many children included in the study (%) had longterm antibiotics given compared to those who did not, and whether there were any differences in the incidence of asthma in the two groups need to be shown.

Minor comments:

1) The abstract conclusion states that "neonatal UTI may have a greater impact on..." however, children who were diagnosed with UTI during the first year of life were included. The authors need to check on the definition of "neonatal".

2) Table 2 mentions "VCUG". It needs to be changed to VUR.

6. PLOS authors have the option to publish the peer review history of their article (what does this mean?). If published, this will include your full peer review and any attached files.

Reviewer #1: No

Reviewer #2: No

---

## [Author Response · Author response to Decision Letter 0]

14 Jun 2021

Dear Editor and Reviewers:

Thank you very much for your insightful comments and extensive suggestions. All your comments we received on this study have been taken into account to improve our manuscript. We have completed the revision based on Reviewers’ comments point by point. The changes were highlighted in the revised manuscript. We hope that these changes to the manuscript will facilitate the decision to publish this study in your journal.

 

Comments to the Author

Reviewer #1: The review for the paper entitled ‘Dose continuous antibiotic prophylaxis for vesicoureteral reflux increase the subsequent risk of childhood asthma?’ has been completed. The authors concluded that the nationwide retrospective cohort study in Taiwan demonstrated that neonatal urinary tract infection (UTI) may have a greater impact on the development of childhood asthma.

However, this paper does not seem to be suitable for publication because of the following reasons:

1) The concept that antibiotic use for various infectious diseases including UTI in early life is associated with the development of later allergic diseases is not novel and there are many articles in this concept (Tanaka M, Nakayama J. Development of the gut microbiota in infancy and its impact on health in later life. Allergol Int. 2017 Oct;66(4):515-522; Gensollen T, Iyer SS, Kasper DL, Blumberg RS. How colonization by microbiota in early life shapes the immune system. Science. 2016 Apr 29;352(6285):539-44; Lin CH, Lin WC, Wang YC, Lin IC, Kao CH. Association Between Neonatal Urinary Tract Infection and Risk of Childhood Allergic Rhinitis. Medicine (Baltimore). 2015 Sep;94(38):e1625; Lin CH, Wang YC, Lin WC, Kao CH. Neonatal urinary tract infection may increase the risk of childhood asthma. Eur J Clin Microbiol Infect Dis. 2015 Sep;34(9):1773-8)

Response: We appreciate your comment. Although the concept that antibiotic use in early life is associated with the development of later allergic diseases is not a novel concept in literature, how subsequent risk of allergic diseases is relating to long-term continuous low dose antibiotics in children with VUR remains unclear. Actually, no RCT or real-life observation study have been performed to examine the difference between long-term continuous low dose antibiotics and short course therapeutic antibiotics on risks of childhood allergic diseases. From Reviewer’s reference, there are some studies investigate the effect of short course of antibiotics on gut microbiota and future risk allergic diseases, but no study examines whether UTI, VUR, or VUR+UTI are really risk factors for asthma. Hence, our study is worth to be conducted. In current study, we also evaluated the incidence and risk of childhood asthma stratified by the type of antibiotics and treatment options for VUR.

We believe our study question is an important issue because approximately 70 % of all the infants who are diagnosed with recurrent febrile urinary tract infection (UTI), will have VUR. Once VUR has been diagnosed, the basic premise in management is to prevent further ascending UTI which would lead to potential renal damage. Since 1960s, antibiotic prophylaxis has been one of the management options in treating VUR to reduce the recurrence rate of UTI in children with VUR, especially in children under age five who may be more susceptible to renal damage by ascending UTI. Actually, some children with low grade VUR may take maintenance antibiotic prophylaxis for several years until such time that the reflux would disappear. To date, most studies concern the issue of long-term antibiotic prophylaxis about the efficiency of reducing frequency of UTI and prevention of renal damage, and antimicrobial resistance with breakthrough infection, but no study investigate whether the use of antibiotic prophylaxis in the management of VUR might be associated with the risk of allergic diseases. 

Reviewer has proposed four articles convey the concept that various infectious diseases including UTI in early life is associated with the development of later allergic diseases, but we believe our study is not the same as previous study and worth to be conducted. The reasons were as follows.

Reference 1: Development of the gut microbiota in infancy and its impact on health in later life. Allergol Int. 2017 Oct;66(4):515-522; Gensollen T, Iyer SS, Kasper DL, Blumberg RS. => In this review article, authors summarize recent findings regarding gut microbiota establishment, including the importance of various factors related to the development of the immune system and allergic diseases later in life. From reviewing this article, there are three studies, which address short-term postnatal antibiotics can have significant effects on the evolution of the infant gut microbiota, the long-term health implications of which remain unknown. (71.Tanaka S, Kobayashi T, Songjinda P, Tateyama A, Tsubouchi M, Kiyohara C, et al. Influence of antibiotic exposure in the early postnatal period on the development of intestinal microbiota. FEMS Immunol Med Microbiol 2009;56:80-7. 72. Fouhy F, Guinane CM, Hussey S, Wall R, Ryan CA, Dempsey EM, et al. High-throughput sequencing reveals the incomplete, short-term recovery of infant gut microbiota following parenteral antibiotic treatment with ampicillin and gentamicin. Antimicrob Agents Chemother 2012;56:5811-20. 73. Greenwood C, Morrow AL, Lagomarcino AJ, Altaye M, Taft DH, Yu Z, et al. Early empiric antibiotic use in preterm infants is associated with lower bacterial diversity and higher relative abundance of enterobacter. J Pediatr 2014;165:23-9.) 

Reference 2: How colonization by microbiota in early life shapes the immune system. Science. 2016 Apr 29;352(6285):539-44 => In this review article, authors discuss the role of early-life education of the immune system during this “window of opportunity,” when microbial colonization has a potentially critical impact on human health and disease. In this review article, authors refer two articles to prove early antibiotics exposure may increase allergic diseases when children grow older. (83. Antibiotic exposure by 6 months and asthma and allergy at 6 years: Findings in a cohort of 1,401 US children. 84. Am J Epidemiol. 2011 Feb 1; 173(3):310-8. 84. Antibiotic use in early life and development of allergic diseases: respiratory infection as the explanation.) One study reports antibiotic use within the first 6 months of life is associated with an increased susceptibility to allergy and asthma at 6 years of age. Another study demonstrates that antibiotic exposure during the first year of life is associated with the development of wheezing and eczema at 8 years of age. However, these 2 studies are observational study, they focus on the association between use of antibiotics in early infancy and subsequent risk of asthma and eczema. But our study is a cohort of children from a nationwide database that were diagnosed with either VUR, UTI, VUR+UTI, and a control group during the first year of life, and investigated the risk of asthma in each of the 4 groups, with the primary endpoint being diagnosed with asthma.

   

Reference 3: Lin CH, Lin WC, Wang YC, Lin IC, Kao CH. Association Between Neonatal Urinary Tract Infection and Risk of Childhood Allergic Rhinitis. Medicine (Baltimore). 2015 Sep;94(38):e1625 

=> This observational study found UTI in newborns is significantly associated with the development of allergic rhinitis in childhood. However, in the previous study, children who were diagnosed with VUR were excluded from the study. 

Reference 4: Lin CH, Wang YC, Lin WC, Kao CH. Neonatal urinary tract infection may increase the risk of childhood asthma. Eur J Clin Microbiol Infect Dis. 2015 Sep;34(9):1773-8) => This observational study found an increased subsequent risk of asthma in children who were diagnosed with neonatal UTI. However, in the previous study, children who were diagnosed with VUR were excluded from the study. 

2) They speculate the association of the history of neonatal UTI with an asthma in later life is caused by dysbiosis of gut microbiota due to antibiotic use in infancy. If so, they should explore the dysbiosis of gut microbiota in neonates with UTI. Without such information, their speculation cannot be supported as numerous confounding factors exist.

Response: We appreciate your comment. We conduct a cohort study of children from a nationwide database that were diagnosed with either VUR, UTI, VUR+UTI, and a control group during the first year of life, and investigated the risk of asthma in each of the 4 groups, with the primary endpoint being diagnosed with asthma. We found that the overall incidence of asthma was highest in the UTI group, and the incidence of asthma was higher in boys than girls. We did not study the association of the history of neonatal UTI with an asthma in later life. The results of previous studies showing dysbiosis of gut microbiota due to antibiotic use in infancy might support or explain our study findings, but such statement is not our conclusion.

  

Reviewer #2: This is a very interesting article on the impact of antibiotic prophylaxis for vesicoureteral reflux and the subsequent risk of childhood asthma. The authors included a cohort of children from a nationwide database that were diagnosed with either VUR, UTI, VUR+UTI, and a control group during the first year of life, and investigated the risk of asthma in each of the 4 groups, with the primary endpoint being diagnosed with asthma. The authors found that the overall incidence of asthma was highest in the UTI group, and the incidence of asthma was higher in boys than girls.

Major comments:

1) Although the author's primary aim of this study is very interesting and the authors have put good efforts to investigate, I believe the statistical analyses which the authors have used to investigate -- comparing the risk of asthma in the selected cohort of patients and finding the HR compared with the control group---is not enough to explain that there is a higher risk of asthma in children with UTI or UTI+VUR. The reason for this is because there are many confounding factors which the authors have not fully adjusted for. I recommend that the authors perform at least a multivariate analyses to find whether UTI, VUR, or VUR+UTI are really risk factors for asthma by adjusting for confounding such as sex, previous antibiotic administration, commodities, types of treatment for VUR...etc. If this is not possible, I believe that the authors can only conclude that there is a higher incidence of asthma in these patient groups; however, whether UTI or VUR or UTI/VUR are risk factors for asthma cannot not be concluded from the data presented in this study.

Response: We appreciate your constructive and insightful comment and we have performed multivariate analyses to find whether UTI, VUR, or VUR+UTI are really risk factors for asthma by adjusting for confounding, including age, sex, urbanization level, and parental occupation, type of antibiotic administration, commodities, types of treatment for VUR, and comorbidities of URI, LRI, otitis media, and sinusitis in revised manuscript (Table 3 and Table 4).

2) The title states that continuous antibiotic prophylaxis for VUR increases the risk of asthma, however, the results and discussion do not support this statement. In fact, the data presented by the authors show that UTI has a significantly higher HR for childhood asthma, and explain that the reason for this may be due to the microorganisms responsible for UTIs which influence the innate immunity triggering the sensitization of asthma. I suggest that the authors change the title to better fit the findings of the study.

Response: We appreciate your insightful comment. We have revised our title to “Association between vesicoureteral reflux, urinary tract infection and antibiotics exposure in infancy and risk of childhood asthma.”

3) The UTI cohort was found to have significantly higher occurrence of respiratory tract infections compared to the VUR/UTI and control cohort. The risk of asthma was highest in the URI cohort, followed by VUR/UTI cohort. Again, with the data that the authors have presented, it is unclear whether UTI is associated with a higher risk of asthma after adjusting for respiratory tract infections.

Response: We appreciate your comment. In the revised manuscript, after we have adjusted comorbidities of URI, LRI, otitis media, and sinusitis, UTI cohort is still associated higher risk for asthma. 

4) In the study participants, patients were included based on whether they had UTI or VUR during the first year of life. Information about whether the study participants had UTI or VUR after the first year of life is important.

Response: We appreciate your insightful comment. Recent studies have begun to define a critical period during early development in which disruption of optimal host-commensal interactions can lead to persistent and in some cases irreversible defects in the development and training of specific immune subsets. The early-life events and antibiotics use might contribute to the development of allergic diseases in later life. In addition, VUR is found in 30–45% of children presenting with a febrile UTI, with an even higher risk in neonates. (Garcia-Roig, M. L., & Kirsch, A. J. (2016). Urinary tract infection in the setting of vesicoureteral reflux.) Hence, based on above concepts, our study is focus on whether UTI, VUR, or VUR+UTI and antibiotics exposure and treatment for VUR within first year are really risk factors for childhood asthma. Therefore, we did not analyze participants had UTI or VUR after the first year of life. Considering your comment, we have added the statement in study limitation. (Page 24, Line 10-14)

5) Table 1 and 2 both have "use of antibiotics". Does table 2 show the type of antibiotics used only to treat UTI's? Did the UTI's occur after the 1st year of life? of after their diagnosis of VUR? These points need to be clarified.

Response: Table 1 shows the number of participants who use antibiotics and Table 2 shows the duration of use antibiotics. In both Table 1 and 2, antibiotics were used for any cause within the study follow-up period. We have written in study results. (Page 13, Line 10-11)

6) For patients without the diagnosis of asthma, what was the age until follow up?

Response: All participants followed up until the asthma diagnosis, their withdrawal from the insurance system (because of death or loss to follow-up), or till the end of 2008. (Page 10, Line 6-8)

7) In the discussion, line 286-288, it is written that "The VUR cohort was exposed to low-dose long-term prophylactic antibiotics but the other two cohorts experienced short-term high-dose therapeutic antibiotics". However, this data is not presented in the results. Data needs to be given on how many children included in the study (%) had longterm antibiotics given compared to those who did not, and whether there were any differences in the incidence of asthma in the two groups need to be shown.

Response: We appreciate your constructive comment. Concept of antibiotic use in early life is associated with the development of later allergic diseases is well-known in literature, but how subsequent risk of allergic diseases is relating to long-term continuous low dose antibiotics remains unclear. To date, no RCT or real-life observation study have been performed to examine the difference between long-term continuous low dose antibiotics and short course therapeutic antibiotics on risks of childhood allergic diseases. 

Approximately 70 % of all the infants who are diagnosed with recurrent febrile urinary tract infection (UTI), will have VUR. Once VUR has been diagnosed, the basic premise in management is to prevent further ascending UTI which would lead to potential renal damage. Since 1960s, antibiotic prophylaxis has been one of the management options in treating VUR to reduce the recurrence rate of UTI in children with VUR, especially in children under age five who may be more susceptible to renal damage by ascending UTI. Actually, some children with low grade VUR may take maintenance antibiotic prophylaxis for several years until such time that the reflux would disappear. To date, most studies concern the issue of long-term antibiotic prophylaxis about the efficiency of reducing frequency of UTI and prevention of renal damage, and antimicrobial resistance with breakthrough infection, but no study investigate whether the use of antibiotic prophylaxis in the management of VUR might be associated with the risk of allergic diseases. 

In this study, VUR only cohort is a group of children who diagnosed as VUR within first year of age, is too young to do surgical intervention to correct VUR. Therefore, they usually take long-term prophylactic antibiotics. Hence, VUR group is a surrogate of participant on long-term prophylactic antibiotics. (Page 10, Line 4-8) 

Minor comments:

1) The abstract conclusion states that "neonatal UTI may have a greater impact on..." however, children who were diagnosed with UTI during the first year of life were included. The authors need to check on the definition of "neonatal".

Response: We have deleted this error word in revised abstract.

2) Table 2 mentions "VCUG". It needs to be changed to VUR.

 Response: We have revised the title of Table 2.

---

## [Decision Letter · Decision Letter 1]

6 Sep 2021

Association between vesicoureteral reflux, urinary tract infection and antibiotics exposure in infancy and risk of childhood asthma

PONE-D-20-27348R1

Dear Dr. Wei,

We’re pleased to inform you that your manuscript has been judged scientifically suitable for publication and will be formally accepted for publication once it meets all outstanding technical requirements.

Kind regards,

Man Ki Kwok

Academic Editor

PLOS ONE

Additional Editor Comments (optional):

Reviewers' comments:

Reviewer's Responses to Questions

**Comments to the Author**

1. If the authors have adequately addressed your comments raised in a previous round of review and you feel that this manuscript is now acceptable for publication, you may indicate that here to bypass the “Comments to the Author” section, enter your conflict of interest statement in the “Confidential to Editor” section, and submit your "Accept" recommendation.

Reviewer #2: All comments have been addressed

2. Is the manuscript technically sound, and do the data support the conclusions?

Reviewer #2: Yes

3. Has the statistical analysis been performed appropriately and rigorously? 

Reviewer #2: Yes

4. Have the authors made all data underlying the findings in their manuscript fully available?

Reviewer #2: Yes

5. Is the manuscript presented in an intelligible fashion and written in standard English?

Reviewer #2: Yes

6. Review Comments to the Author

Reviewer #2: The authors have gone through extensive lengths to make the suggested corrections such as performing multivariate analyses, and it should be commended.

The authors suggested that although the concept that antibiotic use in early life is associated with the development of later allergic diseases is not a novel concept in literature, how subsequent risk of allergic diseases is relating to long-term continuous low dose antibiotics in children with VUR remains unclear, and therefore have undergone this study. They found in the multivariate analyses that in this cohort of children with VUR and UTI do not have a higher risk, therefore, only found a positive correlation with UTI and asthma.

My biggest doubt is that this manuscript conveys that UTI is a risk factor for asthma. However, data shown by the authors can only conclude a positive correlation for a higher incidence of asthma in the UTI group, as said in the conclusion. I suggest softening parts of the manuscript that strongly depict that UTI is a risk factor for asthma, rather, that there is an association of positive correlation.

Minor comments:

Line 360 "Infantile UTI with or without has a positive correlation with the prevalence of..." - is the word "VUR" missing from this sentence?

7. PLOS authors have the option to publish the peer review history of their article (what does this mean?). If published, this will include your full peer review and any attached files.

Reviewer #2: No

---

## [Editor Report · Acceptance letter]

9 Sep 2021

PONE-D-20-27348R1 

Association between vesicoureteral reflux, urinary tract infection and antibiotics exposure in infancy and risk of childhood asthma 

Dear Dr. Wei:

I'm pleased to inform you that your manuscript has been deemed suitable for publication in PLOS ONE. Congratulations! Your manuscript is now with our production department. 

Kind regards, 

on behalf of

Dr. Man Ki Kwok 

Academic Editor

PLOS ONE